# Posted Pricing and Dynamic Prior-independent Mechanisms with Value Maximizers

**Yuan Deng**
Google Research
dengyuan@google.com

**Vahab Mirrokni**
Google Research
mirrokni@google.com

**Hanrui Zhang**
Carnegie Mellon University
hanruiz1@cs.cmu.edu

## Abstract

We study posted price auctions and dynamic prior-independent mechanisms for (ROI-constrained) value maximizers. In contrast to classic (quasi-linear) utility maximizers, these agents aim to maximize their total value subject to a minimum ratio of value per unit of payment made. When personalized posted prices are allowed, posted price auctions for value maximizers can be reduced to posted price auctions for utility maximizers. However, for anonymous posted prices, the well-known $\frac{1}{2}$ approximation for utility maximizers is impossible for value maximizers and we provide a posted price mechanism with $\frac{1}{2}(1 - 1/e)$ approximation. Moreover, we demonstrate how to apply our results to design prior-independent mechanisms in a dynamic environment; and to the best of our knowledge, this gives the first constant revenue approximation with multiple value maximizers. Finally, we provide an extension to combinatorial auctions with submodular / XOS agents.

## 1   Introduction

In online advertising, the growing adoption of autobidding witnesses the emergence of value maximizing bidding, which has become the prevalent behavior model for bidding agents in recent years [Aggarwal et al., 2019, Deng et al., 2021a]. Instead of specifying their bids per auction opportunities, the advertisers only need to report their high-level objectives and/or constraints to the bidding agents and the bidding agents bid on behalf of the advertisers to maximizes their objectives subject to the constraints. A common type of value maximizing bidding is return on investment (ROI)-constrained value-maximizers a.k.a., target CPA (cost per acquisition) and target ROAS (return on ad spend) auto-bidding. For ROI-constrained value-maximizers, their objective is to maximize their total value subject to a constraint specifying a minimum ratio of value per unit of payment made.

In theory, there is already a fairly complete understanding of mechanism design with ROI-constrained value-maximizers. With single-parameter buyers and publicly known target ROI ratios, Balseiro et al. [2021b] show that the VCG auction with properly scaled payments extracts the full optimal welfare as revenue, which is arguably the strongest guarantee one can think of. In order to apply this result, however, there are two major issues:

Firstly, the incentive-compatibility of this optimal mechanism is quite sensitive to the payment scalars, which in turn require prior knowledge to compute. Moreover, when incentive-compatibility is compromised because of (even slightly) inaccurate or misaligned prior beliefs, there is no known way to predict the buyers' behavior, so any guarantee of the mechanism is completely lost. In order to tackle this issue, Balseiro et al. [2021a] propose robust auction formats that are approximately optimal given "signals" that are close enough to the buyers' true values. But what can we do when there is no such signal available? Another recent attempt addresses the prior dependence issues by designing *prior-independent* dynamic auction mechanism with a single ROI-constrained value-maximizer [Deng and Zhang, 2021]. Such a mechanism is useful when the buyer' value distribution is unknown to the seller, and must be learned over time — which is the case in many important application

36th Conference on Neural Information Processing Systems (NeurIPS 2022).

scenarios, such as online ad auctions. Despite significant interest in designing prior-independent dynamic auctions, it remains unknown whether one can even extract a constant fraction of the optimal welfare as revenue in the long run.

Secondly, perhaps an equally important consideration is the cognitive complexity of the mechanism. Despite strong theoretical guarantees it provides, the format of the optimal mechanism (and in particular, the payment scalars) may appear quite mysterious to buyers. As a result, buyers may act suboptimally, and therefore unpredictably, based on their misunderstanding of the mechanism. This can be further exacerbated if incentive-compatiblity is compromised, in which case buyers must come up with their own bidding strategies. All these reasons motivate us to investigate *robust* and *simple* solutions for mechanism design with ROI constraints. In terms of robustness in particular, we are also interested in designing prior-independent mechanisms that do not rely on any kind of predictions.

**Sequential posted price mechanisms.**   In traditional environments, among simple auction formats, the one that receives the most attention is *posted price mechanisms* [Chawla et al., 2010]. Sequential posted price mechanisms are arguably the simplest format of auction protocols (among nontrivial ones): the seller approaches the buyers one by one in an arbitrary order. For each buyer, the seller offers a take-it-or-leave-it price. If the buyer takes the offer, then the buyer gets the item and pays the price, and the auction ends. Otherwise, the seller proceeds to the next buyer and repeats the procedure. In addition to simplicity, posted price mechanisms are also intrinsically robust: with appropriately chosen prices, the guarantees of the mechanism remains approximately valid, even with inaccurate or misaligned prior beliefs. Technically, posted pricing is connected to *prophet inequalities* [Krengel and Sucheston, 1977, 1978], in the sense that the two can be viewed as the same technical problem interpreted in different ways.

**From utility-maximizers to ROI-constrained value-maximizers.**   In traditional settings with utility-maximizers, it is known that in terms of welfare, one can achieve a $(1/2)$-approximation using posted pricing, and this ratio is the best possible.[1] The mechanism used is extremely simple: the seller offers an anonymous price (i.e., same price for all buyers) that is equal to $1/2$ of the expected maximum value across buyers. This guarantee generalizes to multi-unit auctions [Alaei, 2014, Hajiaghayi et al., 2007], and even combinatorial auctions [Dutting et al., 2020, Feldman et al., 2014]. The huge success of posted pricing with utility-maximizers, as well as its simplicity and robustness, brings us to the following natural question: *is it possible to achieve similar guarantees using posted pricing, hopefully with similar pricing strategies, when buyers are ROI-constrained value-maximizers?*

## 1.1   Our Results

In this paper, we initiate the study of posted pricing and prophet inequalities with ROI-constrained value maximizers. The main focus of the paper is on the single-item setting, where $n$ buyers compete for a single indivisible item. We first consider the case of personalized prices, where the seller is allowed to offer a different price for each buyer. We show that with personalized prices, selling to value-maximizers is no harder than selling to traditional utility-maximizers.

**Proposition 1** (Informal Version of Proposition 4)**.** *When personalized prices are allowed, any approximation guarantee in terms of welfare with utility-maximizers implies the same approximation guarantee in terms of revenue against welfare with value-maximizers.*

We then proceed to the more interesting case, where the seller must offer the same, anonymous price to all buyers. Our first result is an upper bound (i.e., impossibility result), which says the usual ratio of $1/2$ is unachievable with an anonymous price, even in terms of welfare, when buyers are ROI-constrained value-maximizers.

**Theorem 1** (Informal Version of Theorem 3)**.** *There exists a problem instance where no anonymous price achieves an approximation ratio better than $0.479$ in terms of welfare.*

Interestingly, the hard instances we present are found by computer-aided search over structured problem instances where the optimal anonymous price can be computed efficiently. Given the upper bound, we move on to the search for a price that achieves a good approximation guarantee, hopefully

---

[1]Essentially the same guarantees can be established for revenue by considering the virtual values.

close to the above upper bound. The most natural candidate is the usual price, $\frac{1}{2}\mathbb{E}[\max_i v_i]$ (where $v_i$ is buyer $i$'s value), that has been extensively studied in posted pricing and prophet inequalities with utility-maximizers. This price and its generalizations achieve the optimal ratio of $1/2$ in most natural settings with utility-maximizers. While this is no longer possible give the upper bound, we show this price still achieves a decent approximation ratio even with value-maximizers. And in fact, the ratio given by our analysis is the best possible for this price.

**Theorem 2** (Informal Version of Theorem 4 and Proposition 5). *For any problem instance, offering the price of $\frac{1}{2}\mathbb{E}[\max_i v_i]$, where $v_i$ is buyer $i$'s value, to all buyers extracts a $\frac{1}{2}(1 - 1/e) \approx 0.316$ fraction of the optimal welfare as revenue. Moreover, our analysis is tight for this price.*

Finally, we demonstrate the wide applicability of our techniques by showing how they can be useful in two related problems: prior-independent dynamic auctions and combinatorial auctions with value-maximizers. For prior-independent dynamic auctions, we prove the following result.

**Proposition 2** (Informal Version of Proposition 6). *There is a prior-independent dynamic auction mechanism that extracts a $\frac{1}{2}(1 - 1/e)$ fraction of the optimal welfare as revenue in the long run.*

To our knowledge, this is the first nontrivial revenue guarantee for prior-independent dynamic mechanism with multiple value-maximizers (the case with a single buyer has been studied very recently [Deng and Zhang, 2021]). For combinatorial auctions, through an alternative analysis of the usual price, we prove the following result.

**Proposition 3** (Informal Version of Proposition 7). *In combinatorial auctions with value-maximizers, there are anonymous item prices that achieve an approximation ratio of $1/4$ in terms of welfare.*

To our knowledge, this is the first nontrivial result for combinatorial auctions with value-maximizers.

## 1.2 Further Related Work

**Mechanism design with value-maximizers.** Aggarwal et al. [2019] initiate the study of ROI-constrained value maximizers and show that VCG mechanism can achieve at most 1/2 of the optimal social welfare in the worst case, which inspire a series of follow-up works to find ways to improve the approximation ratio. Balseiro et al. [2021a] and Deng et al. [2021a] demonstrate that with machine learning advice that approximates the advertisers' values well, the mechanism design can use boosts and/or reserves based on the advice to improve the efficiency guarantees. Balseiro et al. [2021b] design revenue-optimal mechanisms under various information structures in the Bayesian setting. Deng and Zhang [2021] design prior-independent mechanisms in an online environment by leveraging the structure of the optimal mechanism from Balseiro et al. [2021b].

**Posted pricing and prophet inequalities.** Prophet inequalities were initially introduced in the context of optimal stopping theory [Krengel and Sucheston, 1977, 1978], and later re-introduced to the CS community by Hajiaghayi et al. [2007]. Since then, its connection to posted pricing has been extensively studied and exploited. For a detailed exposition on the connection between prophet inequalities and posted pricing, see the survey by Lucier [2017]. In the past two decades, posted pricing and prophet inequalities have proved useful in an extremely wide range of settings, from simple single-parameter settings [Azar et al., 2014, Correa et al., 2019a,b, Dütting and Kesselheim, 2019, Hajiaghayi et al., 2007, Rubinstein et al., 2020], to matroid and knapsack constraints [Caramanis et al., 2022, Chawla et al., 2010, Dutting et al., 2020, Ehsani et al., 2018, Kleinberg and Weinberg, 2012], to general feasibility constraints [Rubinstein, 2016], to combinatorial objective functions [Rubinstein and Singla, 2017], to simple multi-parameter settings [Chawla et al., 2010], to combinatorial auctions with submodular/XOS [Dutting et al., 2020, Ehsani et al., 2018, Feldman et al., 2014] and subadditive valuations [Dütting et al., 2020, Zhang, 2022]. Similar techniques have also proved useful in online settings [Cohen et al., 2014, Deng et al., 2021b]. All these results are under the traditional assumption of utility-maximizing agents. In contrast, we consider posted pricing with value-maximizers, which, as we will see, creates significant differences and new challenges, both conceptuallly and technically.

## 2 Preliminaries

**Basic setup.** We consider selling a single indivisible item to $n$ buyers. Each buyer $i$ has a value $v_i$ drawn independently from a distribution $D_i$. For simplicity, unless otherwise specified, we always

assume each $D_i$ is non-atomic, i.e., the CDF of $D_i$ is continuous, although all our results still apply without the assumption. We focus on posted price mechanisms in this paper, where the seller chooses a price $p_i$ for each buyer $i$ based on the value distributions $\{D_i\}_i$. The buyers then arrive in an adversarial order. Upon the arrival of buyer $i$, if $i$ decides to accept the price, then the seller's revenue is $p_i$, and the auction ends. Otherwise, the next buyer arrives, and decides whether to accept the price, etc. If no buyer accepts their price, then the seller's revenue is $0$.

**ROI-constrained value-maximizers.**  Now we describe how ROI-constrained value-maximizing buyers decide whether to accept a price. Without loss of generality, we assume each buyer's target ROI ratio is $1$. Each buyer's goal is to maximize their expected value, subject to the constraint that the expected payment cannot exceed the expected value. This is captured by the following program.

$$\text{maximize} \quad \mathbb{E}_{v \sim D}[x(v) \cdot v]$$
$$\text{subject to} \quad \mathbb{E}_{v \sim D}[x(v) \cdot v \geq x(v) \cdot p],$$

where $D$ is the buyer's value distribution, $p$ is the price, and the variable $x : \mathbb{R}_+ \to \{0, 1\}$ is the buyer's strategy mapping the realized value $v$ to "accept" (i.e., $1$) or "reject" (i.e., $0$). Conceptually, this corresponds to settings where auctions happen repeatedly, and the buyer cares about the cumulative value and payment in the long run. It is not hard to show that the optimal solution to the above program is

$$x(v) = \begin{cases} 1, & \text{if } v \geq \theta(D, p) \\ 0, & \text{otherwise} \end{cases},$$

where

$$\theta(D, p) = \inf\{\theta \in \mathbb{R}_+ \mid \mathbb{E}_{v \sim D}[v \mid v \geq \theta] \geq p\}.$$

For consistency we say $\inf \emptyset = \infty$. So, a buyer with value distribution $D$ facing a price $p$ accepts the price, iff the realized value $v$ is greater than or equal to $\theta(D, p)$.

**Seller's objective: revenue maximization.**  Following conventions in mechanism design with ROI-constrained value-maximizers, we assume the seller's objective is to maximize expected revenue. Moreover, the benchmark that we compare to is the maximum expected welfare, i.e., $\mathbb{E}_{\{v_i\} \sim \{D_i\}}[\max_i v_i]$. Our goal is to maximize the ratio between the seller's expected revenue and the maximum expected welfare. Note that since buyers are ROI-constrained, any revenue guarantee immediately implies a welfare guarantee of the same factor.

## 3 Warm-up: Posted Pricing with Personalized Prices

We first consider the case where personalized prices are allowed, i.e., for two buyers $i_1$ and $i_2$, the prices offered by the seller, $p_{i_1}$ and $p_{i_2}$, are not necessarily the same. We show that with personalized prices, any guarantee that is achievable in traditional settings with utility-maximizers is also achievable with ROI-constrained value-maximizers. The proof is fairly simple, but reveals key connections and differences between utility-maximizers and ROI-constrained value-maximizers, which will be instrumental in our later discussion. Formally, we prove the following claim.

**Proposition 4.** *For any number of buyers $n$ and value distributions $D_1, \ldots, D_n$, there exist personalized prices $p_1, \ldots, p_n$, such that the seller's expected revenue is at least $\frac{1}{2} \mathbb{E}_{\{v_i\} \sim \{D_i\}}[\max_i v_i]$.*

*Proof.* We present a reduction to posted pricing with utility-maximizers. That is, given prices that guarantee an $\alpha$-approximation in terms of welfare with utility-maximizers, we construct prices that extract an $\alpha$ fraction of the maximum welfare as revenue with ROI-constrained value-maximizers. The proposition follows immediately since there are known $1/2$-approximation prices with utility-maximizers.

Consider any prices $q_1, \ldots, q_n$ for utility-maximizers with value distributions $D_1, \ldots, D_n$. Without loss of generality, we also assume each $q_i$ is in the support of $D_i$. We construct prices $p_1, \ldots, p_n$ that induce exactly the same allocation with ROI-constrained value-maximizers for every combination of realized values, as that induced by $q_1, \ldots, q_n$ with utility-maximizers. For each $i$, let $p_i$ be such that $\theta(D_i, p_i) = q_i$ (this is always possible since $q_i$ is in the support of $D_i$). Observe that the behavior of

a utility-maximizer facing price $q_i$ is the same as that of an ROI-constrained value-maximizer facing price $p_i$. In the former case, the buyer accepts the price iff the value $v_i \geq q_i$. In the latter case, the buyer accepts the price iff the value $v_i \geq \theta(D_i, p_i)$, which is equal to $q_i$.

Given the above, we immediately see that the welfare guaranteed by $p_1, \ldots, p_n$ with ROI-constrained value-maximizers is the same as that guaranteed by $q_1, \ldots, q_n$ with utility-maximizers. We only need to argue that the revenue guaranteed by $p_1, \ldots, p_n$ is the same as the welfare. To this end, observe that the ROI constraint is binding for every buyer $i$. That is, the expected value of each buyer $i$ is equal to the expected payment the buyer makes. This may appear trivial given the definition of $\theta(D, p)$, but actually it is not: consider a buyer whose value is constantly 10. When facing a price of 1, the buyer always accepts the price, but clearly the value is much higher than the payment. Nevertheless, the two are always equal if the price is at least the expected value of the buyer, i.e., when $p \geq \mathbb{E}_{v \sim D}[v]$. This is because in such cases, there exists a $\theta$ such that $\mathbb{E}_{v \sim D}[v \mid v \geq \theta] = p$, which by definition implies $\mathbb{E}_{v \sim D}[v \mid v \geq \theta(D, p)] = p$. Our construction does satisfy this condition.[2] Now summing over the binding ROI constraints, we immediately see that the revenue is equal to the welfare, which concludes the proof. $\qquad\square$

Another way to interpret Proposition 4 is the following: one can consider the Lagrangified version of each buyer's decision problem. Suppose the optimal Lagrange multiplier is $\lambda^*$. Observe that if $q = \frac{p \cdot \lambda^*}{1 + \lambda^*}$, then the problem of a value-maximizer facing price $p$ is the same as the problem of a utility-maximizer facing price $q$. This also gives a way of constructing prices $p_1, \ldots, p_n$ for value-maximizers based on existing prices $q_1, \ldots, q_n$ for utility-maximizers.

We make two remarks regarding the above reasoning.

- The new prices $p_1, \ldots, p_n$ in general are different even if the old ones $q_1, \ldots, q_n$ are the same. This is because each $p_i$ also depends on $D_i$, in addition to $q_i$. So, the existence of an anonymous price that guarantees $1/2$ of the optimal welfare with utility-maximizers does not imply the same guarantee with ROI-constrained value-maximizers using an anonymous price. In fact, as we will show later, with ROI-constrained value-maximizers, it is impossible to achieve the ratio of $1/2$ using an anonymous price.

- With ROI-constrained utility maximizers, the "interesting" case is when all ROI constraints are binding. This is because if some buyer's ROI constraint is not binding, then that buyer must always accept the price, which means the revenue of the seller is at most the price for that buyer (when that buyer arrives first). Restricted to the case where all ROI constraints are binding, the revenue of the seller is always equal to the welfare, and it may sometimes help to reason about the latter, as we will see.

## 4 Posted Pricing with an Anonymous Price

As Proposition 4 shows, posted pricing with ROI-constrained value-maximizers is easy with personalized prices, but for various practical reasons we may want a single anonymous price for all buyers. In that case, the reduction approach of Proposition 4 fails completely. In this section, we present our results on posted pricing with an anonymous price, which also involve some intriguing technical ingredients.

### 4.1 An Upper Bound Strictly below $0.5$

Our first result is an upper bound on the approximation ratio, which says it is impossible to achieve the familiar ratio of $1/2$ using an anonymous price when buyers are ROI-constrained value-maximizers.

**Theorem 3.** *With $n = 4$ buyers, there exist value distributions $D_1, \ldots, D_4$, such that no anonymous price extracts more than $0.483$ of the optimal welfare as revenue. With $n = 5$ buyers, the ratio further degrades to $0.479$. Moreover, the same lower bounds apply even if we optimize for the welfare.*

---

[2]Recall that we require $q_i$ to be in the support of $D_i$ in (this is without loss of generality, because if $q_i$ is not in the support, we can increase it in a way that the probability that the buyer accepts $q_i$ stays the same, until $q_i$ is back in the support). Then we can choose $p_i$ such that $\theta(D_i, p_i) = q_i$, and $p_i$ must be unique since we also assume $D_i$ is non-atomic, which also means $\mathbb{E}[v_i \mid v_i \geq q_i] = p_i$. On the other hand, we know that $\mathbb{E}[v_i \mid v_i \geq x]$ increases monotonically in $x$, and $q_i \geq 0$, so $p_i = \mathbb{E}[v_i \mid v_i \geq q_i] \geq \mathbb{E}[v_i \mid v_i \geq 0] = \mathbb{E}[v_i]$.

The proof of the theorem, as well as all other missing proofs, is deferred to the appendix. Interestingly, the hard instances we present are found by computer-aided search over structured problem instances. To be more specific, we consider "binary" value distributions, where the value of each buyer $i$ is either some positive number $y_i$ or $0$. The optimal welfare for such instances is easy to compute: we simply sort all buyers in decreasing order of $y_i$ and allocate to the first buyer whose value realizes into $y_i$ (rather than $0$). On the other hand, the optimal anonymous price can also be efficiently computed: in fact, we show that the price is (without loss of generality) equal to $y_i$ for some buyer $i$, so to compute the optimal price we only need to try all $y_i$'s. We then obtain the upper bound by generating random instances with binary value distributions and computing the optimal welfare and the optimal revenue from an anonymous price, respectively.

## 4.2 Approximation Guarantee of the Usual Price

Now we present the main technical result of the paper, which states that the usual price of $\frac{1}{2}\mathbb{E}[\max_i v_i]$ extracts at least $\frac{1}{2}(1-1/e)$ of the optimal welfare as revenue. Formally, we prove the following result.

**Theorem 4.** *Fix any number of buyers $n$ and value distributions $D_1, \ldots, D_n$. With ROI-constrained value-maximizing buyers, when the seller offers an anonymous price of $p = \frac{1}{2}\mathbb{E}_{\{v_i\}\sim\{D_i\}}[\max_i v_i]$ to every buyer, the resulting revenue is at least*

$$\frac{1}{2}\left(1 - \frac{1}{e}\right) \cdot \mathbb{E}_{\{v_i\}\sim\{D_i\}}\left[\max_i v_i\right].$$

To prove Theorem 4, we only need to show that with probability at least $1 - 1/e$, at least one buyer accepts the price $p$. We do this by constructing another price $p'$ satisfying (1) $p' \geq p$, and (2) with probability at least $1 - 1/e$, at least one buyer accepts $p'$. Formally, the proof of Theorem 4 relies on the following lemma.

**Lemma 1.** *Fix any number of buyers $n$ and value distributions $D_1, \ldots, D_n$. Let $p'$ be the largest real number such that*

$$\sum_{i\in[n]} \Pr_{v_i\sim D_i}[v_i \geq \theta(D_i, p')] = 1.$$

*Then $p'$ satisfies*

$$p' \geq \frac{1}{2}\mathbb{E}_{\{v_i\}\sim\{D_i\}}\left[\max_i v_i\right].$$

*And moreover, with probability at least $1 - 1/e$, at least one buyer accepts $p'$, i.e.,*

$$1 - \prod_i(1 - \Pr_{v_i\sim D_i}[v_i \geq \theta(D_i, p')]) \geq 1 - \frac{1}{e}.$$

Here we give a sketch of the proof of the lemma. First observe that by the choice of $p'$, the sum of the probabilities that each buyer $i$ accepts the price $p'$ is 1. By independence and concavity, the probability that at least one buyer accepts $p'$ must be at least $1 - 1/e$. The harder part is to lower bound $p'$ by $\frac{1}{2}\mathbb{E}[\max_i v_i]$. To this end, we compare against an "ex-ante relaxation" of $\mathbb{E}[\max_i v_i]$: for each $i$, we let $\alpha_i$ be the probability that $v_i$ is the largest among all realized values, and let $\beta_i$ be the top $\alpha_i$ quantile of $D_i$ (i.e., the probability that $v_i \geq \beta_i$ is precisely $\alpha_i$). Then one can show that the sum (over $i$) of the contribution to $\mathbb{E}[v_i]$ above $\beta_i$ (i.e., $\alpha_i$ times the conditional expectation of $v_i$ given $v_i \geq \beta_i$) is an upper bound for $\mathbb{E}[\max v_i]$. So we only need to compare $p'$ against this sum. Here, we partition the sum into two parts: the contribution of buyers $i$ where $\beta_i \geq \theta(D_i, p')$, and the contribution of buyers $i$ where $\beta_i < \theta(D_i, p')$. We argue that $p'$ is at least as large as the larger one between the two parts, which gives the factor of $\frac{1}{2}$. We then give two different arguments for comparison against the two parts respectively, which rely on a combination of properties of $\theta(\cdot, \cdot)$, $p'$, and the ex-ante relaxation.

Once we have Lemma 1, it is not hard to prove Theorem 4.

*Proof of Theorem 4.* Observe that the probability that at least one buyer accepts the price is non-increasing in the price. Now by Lemma 1, our price $p$ in Theorem 4 is no larger than $p'$ in Lemma 1.

So the probability that at least one buyer accepts our price $p$ is no smaller than the probability that at least one buyer accepts $p'$, and again by Lemma 1, the latter probability is at least $1 - 1/e$. So the revenue extracted by offering $p$ is at least

$$\left(1 - \frac{1}{e}\right) p = \frac{1}{2} \left(1 - \frac{1}{e}\right) \cdot \mathop{\mathbb{E}}_{\{v_i\} \sim \{D_i\}} \left[\max_i v_i\right]. \qquad \square$$

**Tightness of analysis.** Given the seemingly unnatural factor of $\frac{1}{2}(1 - 1/e)$, one may wonder if our analysis of the price $p$ is tight. The following result shows it in fact is.

**Proposition 5.** *For any $c > 0$, there exists $n$ and $D_1, \ldots, D_n$, such that offering the price $p = \frac{1}{2} \mathbb{E}[\max_i v_i]$ extracts revenue at most*

$$\frac{1}{2} \left(1 - \frac{1}{e} + c\right) \cdot \mathbb{E}\left[\max_i v_i\right].$$

Here we sketch the problem instances used to prove tightness. There is a single "safe" buyer, whose value is always some fixed number (say $k$). In addtion, there are about $k$ "risky" buyers, each of which has value $1/\varepsilon$ with probability $\varepsilon$, where $\varepsilon$ is a small positive number. The expected optimal welfare is about $2k$, so the price we post is about $k$. We can perturb the numbers so that the price is a bit higher than the value of the safe buyer, and that buyer never accepts the price. Now the only source of revenue is the risky buyers. Since the expected value of each risky buyer is about $1$, each of them accepts the price of about $k$ with probability about $1/k$, and the probability that at least one of them accepts the price is about $1 - 1/e$. So, the revenue (and welfare) from posting $\frac{1}{2} \mathbb{E}[\max v_i]$ in this instance is about $(1 - 1/e)k$, whereas the optimal welfare is about $2k$. The ratio matches the bound we prove in Theorem 4.

**Remark on robustness.** Finally, we remark that posted pricing can in fact be robust even with ROI-constrained value-maximizers. One simple way to guarantee robustness is to slightly lower the price offered, by an amount proportional to how inaccurate or misaligned the prior beliefs can be (which of course requires an appropriate measure of inaccuracy). Then, it is not hard to argue that the probability that at least one buyer accepts the price is as expected, even with inaccurate or misaligned prior beliefs. Any possible loss in revenue is therefore only from slightly lowering the price.

## 5 Prior-Independent Dynamic Auctions with Value-Maximizers

In this and the following section, we discuss further implications and generalizations of our results, which demonstrate the power of the posted pricing framework with ROI-constrained value-maximizers.

One important question in auction design with autobidders is whether there exists a no-regret prior-independent dynamic auction mechanism with ROI-constrained value-maximizers. In many practical applications such as online ad auctions, the buyers' value distributions are unknown to the seller, and must be learned over time. Deng and Zhang [2021] give such a mechanism when there is only one buyer, but the case with multiple buyers remain open. Below we show how our results imply a partial answer to this question: there exists a prior-independent dynamic auction mechanism that in the long run, extracts a constant fraction of the optimal welfare as revenue.

**Setup.** The dynamic environment we consider is similar to that studied in [Deng and Zhang, 2021]. Below we only give an informal description of the environment (see [Deng and Zhang, 2021] for more details). Compared to the static setting considered above, in the dynamic setting, auctions happen repeatedly over time. Each buyer's value distribution remains the same throughout the entire procedure. In each time period, each buyer draws a new value independently from their own value distribution, and each time period has its own ROI constraints. We require the mechanism to be prior-independent, which means it cannot depend on the value distributions (but can depend on historical observations of the buyers' behavior). We also assume the value distributions are supported on $[0, 1]$, which is a common assumption in prior-independent auctions.

**A bi-criteria mechanism via posted pricing.** We present a dynamic mechanism that extracts a $\frac{1}{2}(1 - 1/e)$ fraction of the optimal welfare in the long run. We do this by reducing the problem to

no-regret learning the optimal anonymous price: in each time period, we run a sequential posted price auction with an anonymous price, which is chosen using any off-the-shelf algorithm for finite-armed stochastic bandits[3] after discretization. Formally, we prove the following.

**Proposition 6.** *With ROI-constrained value-maximizing buyers, there is a prior-independent dynamic mechanism that, for any number of ROI-constrained value-maxmizing buyers $n$, value distributions $D_1, \ldots, D_n$ and time horizon $T$, extracts revenue at least*

$$\frac{1}{2}\left(1 - \frac{1}{e}\right) \cdot \mathop{\mathbb{E}}_{\{v_i\} \sim \{D_i\}}\left[\max_i v_i\right] \cdot T - O(T^{2/3}).$$

We remark that if buyers care about the future (i.e., they have a positive discount factor, as studied in [Amin et al., 2014, Babaioff et al., 2009, Deng and Zhang, 2021, Nedelec et al., 2022]), then they may still have incentives to lie in response to the above mechanism. However, as long as buyers are less patient than the seller, it is not hard to design a dynamic mechanism based on our posted-price mechanism, where even patient buyers have no incentive to lie. For example, one can adapt the exploration-exploitation framework in [Deng and Zhang, 2021] in the following way: we first run the exploration mechanism in [Deng and Zhang, 2021] for each buyer for sufficiently many time periods to learn the approximate value distributions of all buyers. Then we run our posted-price mechanism with the price slightly lowered to account for potential inaccuracy in the value distributions learned earlier. By trading off between the lengths of the exploration phase and the exploitation phase, one can achieve regret $\tilde{O}(T^{2/3})$ against a $(1 - 1/e)/2$ fraction of the optimal revenue.

## 6 Combinatorial Auctions with Value-Maximizers

With utility-maximizers, posted pricing schemes generalize elegantly to combinatorial auctions, where multiple heterogeneous, possibly mutually substituting, items are sold. One may naturally wonder if similar generalizations exist with ROI-constrained value-maximizers. We demonstrate one way to generalize our results to combinatorial auctions with submodular or XOS valuations. In exchange for generality, we get a worse approximation factor of $1/4$, which applies to welfare but not revenue. To our knowledge, this is the first mechanism that achieves nontrivial guarantees in combinatorial auctions with ROI-constrained value-maximizers.

**Setup.** The setup we consider is similar to that studied in [Feldman et al., 2014], except that we consider ROI-constrained value-maximizers instead of utility-maximizers. There are $m$ heterogeneous items, and each buyer $i$ has a valuation function $v_i : [m] \to \mathbb{R}_+$, drawn independently from $i$'s valuation distribution $D_i$. Following prior research on combinatorial auctions, we assume each buyer $i$'s valuation function $v_i$ is submodular or XOS (we only use certain properties of these classes in a blackbox way; for formal definitions see, e.g., [Feldman et al., 2014]). Such functions model items that are potentially substitutes, but never complements, to each other. We consider posted price mechanisms, in which each item $j \in [m]$ is associated with an anonymous price $p_j$. Buyers arrive in an adversarial order. Upon arrival, each buyer $i$ can choose to buy any subset of the items that are still available, and the total payment $i$ pays is the sum of the prices of the items bought. Once sold to a buyer, an item immediately becomes unavailable.

**Buyer's problem.** Here, we deviate from the setup introduced in Section 2, and instead consider ROI constraints over different items. Each buyer $i$'s ROI constraint is over all items that $i$ receives and the total payment that $i$ makes. That is, when $i$ receives items $S \subseteq [m]$ and pays $p$ in total, the ROI constraint requires that $v_i(S) \geq p$. So, when a buyer has valuation function $v$, the set of available items is $A$, and the prices are $\{p_j\}_{j \in A}$, the buyer's problem is captured by the following program.

$$\begin{aligned} \text{maximize} \quad & v(S) \\ \text{subject to} \quad & v(S) \geq \sum_{j \in S} p_j, \end{aligned}$$

where the variable $S \subseteq A$ is the set of items that the buyer buys. We let $\mathrm{BUY}(v, A) \subseteq A$ denote the optimal solution to the above program. We allow the buyer to break ties arbitrarily. We also note that

---

[3]To achieve the claimed regret bound, one may run Thompson Sampling [Bubeck and Liu, 2013, Thompson, 1933] or certain versions of UCB [Auer et al., 2002, Lattimore and Szepesvári, 2020]).

in the limit, this setup generalizes the single-item setup introduced in Section 2: when each buyer's valuation function is additive, and the value of each item is iid, we effectively recover the single-item setup by letting $m \to \infty$.

**The mechanism.** The mechanism we analyze is exactly the same as the one proposed in [Feldman et al., 2014]. Let $\text{OPT}_i(v_1, \ldots, v_n)$ be the set of items that buyer $i$ receives in the welfare-maximizing allocation, when the valuation functions are $v_1, \ldots, v_n$. We use the following property (see, e.g., [Dutting et al., 2020, Feldman et al., 2014]) of submodular and XOS valuations.

**Lemma 2.** *Fix any XOS valuation $v$ and set of items $S \subseteq [m]$. There exist nonnegative numbers $\{a_j\}_{j \in S} = \{a_j(v, S)\}_{j \in S}$ such that (1) $\sum_{j \in S} a_j = v(S)$, and (2) for any $T \subseteq S$, $\sum_{j \in T} a_j \leq v(T)$.*

We also remark that these numbers can be computed efficiently with oracle access to the valuation function (see [Dutting et al., 2020]). Given this property, for each item $j$, the price we pick is

$$p_j = \frac{1}{2} \mathop{\mathbb{E}}_{\{v_i\} \sim \{D_i\}} \left[ \sum_i a_j(v_i, \text{OPT}_i(v_1, \ldots, v_n)) \right],$$

where we let $a_j(v, S) = 0$ if $j \notin S$. Intuitively, this is setting each item's price to half of its expected contribution to the maximum welfare. These prices generalize the one in the single-item setting. We prove the following guarantee of these prices.

**Proposition 7.** *For any $n$, $m$, and valuation distributions $D_1, \ldots, D_n$, there exist anonymous prices $p_1, \ldots, p_m$ which guarantee expected welfare at least*

$$\frac{1}{4} \mathop{\mathbb{E}}_{\{v_i\} \sim \{D_i\}} \left[ \sum_i v_i(\text{OPT}_i(v_1, \ldots, v_n)) \right].$$

The proof of Proposition 7 is similar to the analysis of the same mechanism for utility-maximizers (see, e.g., [Feldman et al., 2014]). The key difference is that with value-maximizers, the welfare is no longer equal to the sum of the revenue and buyers' utility. Instead, we only have the weaker guarantee that the welfare is at least as large as the larger one between the revenue and buyers' utility, which is at least as large as $1/2$ of the sum of the two. Here we lose a factor of 2.

# 7 Conclusion and Future Work

In this paper, we initiate the study of posted pricing and prophet inequalities with ROI-constrained value-maximizers. We show that with personalized prices, posted pricing with value-maximizers is no harder than with traditional utility-maximizers. For the more interesting case of pricing with an anonymous price, we give nontrivial upper and lower bounds. In particular, our lower bound is through a tight analysis of the usual threshold of $\frac{1}{2} \mathbb{E}[\max_i v_i]$, and our upper bound is strictly below $1/2$. The most natural open question is to determine the optimal ratio with an anonymous price. We also show how our techniques can be applied to two related problems: prior-independent dynamic auctions and combinatorial auctions with value-maximizers. To this end, future directions also include improving the approximation guarantees for these problems, as well as further generalizing to other related problems.

## Acknowledgments and Disclosure of Funding

We thank anonymous reviewers for their helpful feedback.

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
