## A Omitted Proofs

*Proof of Theorem 3.* We only need to prove the claim for the welfare as the objective, since it upper bounds the revenue. We consider buyers with binary value distributions. That is, each buyer $i$ can be described by two quantities: $x_i$ is the probability that the buyer has a positive value, and $y_i$ is the value of the buyer when it is positive. Without loss of generality, we assume $y_i \geq y_{i+1}$ for each $i \in [n-1]$. We also assume a fixed order of arrival, i.e., $1, 2, \ldots, n$. This can only make the problem easier.

Observe the following nice property: if all buyers have binary distributions, then without loss of generality the optimal anonymous price is equal to some $y_i$. To see why this is the case, consider a price $p \in (y_{k+1}, y_k]$ for some $k \in [n]$ (where we let $y_{n+1} = 0$). We argue that the welfare achieved by offering $p$ cannot be larger than that achieved by offering a price of $y_k$. We only need to consider the first $k$ buyers, since buyers $k+1, \ldots, n$ never accept the price $p$. For each $i \in [k]$, the probability $r_i$ that $i$ accepts the price is

$$r_i = \begin{cases} 1, & \text{if } x_i y_i \geq p \\ x_i y_i / p, & \text{otherwise} \end{cases}.$$

Observe (1) $r_i$ is non-decreasing in $p$, and (2) $r_i \geq x_i$ for each $i \leq k$, since $y_i \geq y_k \geq p$. So each buyer $i \leq k$ accepts the price whenever their value is positive, and sometimes accepts the price even if their value is 0.[4] In particular, if the auction has not ended when buyer $i$ arrives, then $i$ contributes the entire expected value $x_i y_i$ to the welfare. The welfare achieved by offering $p$ is therefore

$$\sum_{i \in [k]} \prod_{i' < i} (1 - r_{i'}) x_i y_i.$$

This is non-increasing in $p$ on $(y_{k+1}, y_k]$. So offering $p$ cannot be better than offering $y_k$.

Now we are ready to present the hard instances. For $n = 4$, consider the following binary distributions:

- $x_1 = 0.013344$, $y_1 = 0.683396$,

---

[4]With binary distributions a buyer's best response is generally not unique. Here we consider the one that maximizes the probability of accepting the price, which can only make the problem easier. Also note that this non-uniqueness cannot happen when value distributions are non-atomic.

- $x_2 = 0.014717$, $y_2 = 0.547734$,

- $x_3 = 0.316807$, $y_3 = 0.024032$,

- $x_4 = 0.986649$, $y_4 = 0.017271$.

One may check that the optimal welfare is about $0.035792$. On the other hand, the welfare achieved by offering $y_1$, $y_2$, $y_3$, and $y_4$ respectively is about $0.009119$, $0.017046$, $0.017261$, and $0.017239$. This gives a ratio of about $0.482266$. For $n = 5$, one may consider the following distributions:

- $x_1 = 0.006271$, $y_1 = 0.926251$,

- $x_2 = 0.014677$, $y_2 = 0.629891$,

- $x_3 = 0.167392$, $y_3 = 0.026910$,

- $x_4 = 0.447439$, $y_4 = 0.017906$,

- $x_5 = 0.937673$, $y_5 = 0.015669$.

One may check this gives a ratio of about $0.478595$. $\qquad\square$

*Proof of Lemma 1.* We first prove the easier part, the second bullet point. Observe that $\log(1 - x)$ is concave on $[0, 1)$, and as a result,

$$1 - \prod_i \left(1 - \Pr_{v_i \sim D_i}[v_i \geq \theta(D_i, p')]\right) \geq 1 - \left(1 - \frac{1}{n} \sum_i \Pr_{v_i \sim D_i}[v_i \geq \theta(D_i, p')]\right)^n$$
$$= 1 - (1 - 1/n)^n \qquad \text{(choice of } p')$$
$$\geq 1 - 1/e.$$

Now we prove the first bullet point. We consider an ex-ante relaxation of the optimal welfare. For each $i \in [n]$, let $\alpha_i$ be the probability that $v_i$ is the highest value among all buyers, i.e.,

$$\alpha_i = \Pr_{\{v_{i'}\} \sim \{D_{i'}\}}[v_i = \max_{i'} v_{i'}].$$

Clearly we have

$$\sum_i \alpha_i = 1.$$

Moreover, for each $i$, let $\beta_i$ be the threshold such that

$$\Pr_{v_i \sim D_i}[v_i \geq \beta_i] = \alpha_i.$$

Then we have

$$\sum_i \alpha_i \cdot \mathbb{E}_{v_i \sim D_i}[v_i \mid v_i \geq \beta_i] = \sum_i \Pr_{v_i \sim D_i}[v_i \geq \beta_i] \cdot \mathbb{E}_{v_i \sim D_i}[v_i \mid v_i \geq \beta_i]$$
$$\geq \sum_i \Pr_{\{v_{i'}\} \sim \{D_{i'}\}}\left[v_i = \max_{i'} v_{i'}\right] \cdot \mathbb{E}_{\{v_{i'}\} \sim \{D_{i'}\}}\left[v_i \mid v_i = \max_{i'} v_{i'}\right]$$
$$= \mathbb{E}_{\{v_i\} \sim \{D_i\}}\left[\max_i v_i\right].$$

Now we partition $[n]$ into two sets:

$$A = \{i \in [n] \mid \beta_i \geq \theta(D_i, p')\} \quad \text{and} \quad B = \{i \in [n] \mid \beta_i < \theta(D_i, p')\}.$$

The plan is to show

$$p' \geq \max\left\{\sum_{i \in A} \alpha_i \cdot \mathbb{E}_{v_i \sim D_i}[v_i \mid v_i \geq \beta_i], \sum_{i \in B} \alpha_i \cdot \mathbb{E}_{v_i \sim D_i}[v_i \mid v_i \geq \beta_i]\right\}.$$

Consider $A$ first. Observe that for each $i \in A$,

$$\Pr_{v_i \sim D_i}[v_i \geq \theta(D_i, p')] \cdot \mathbb{E}_{v_i \sim D_i}[v_i \mid v_i \geq \theta(D_i, p')] \geq \alpha_i \cdot \mathbb{E}_{v_i \sim D_i}[v_i \mid v_i \geq \beta_i].$$

This is simply because

$$\Pr_{v_i \sim D_i}[v_i \geq x] \cdot \mathbb{E}_{v_i \sim D_i}[v_i \mid v_i \geq x]$$

is non-increasing in $x$. On the other hand, by the choice of $p'$ and the definition of $\theta(\cdot, \cdot)$,

$$p' = \mathbb{E}_{v_i \sim D_i}[v_i \mid v_i \geq \theta(D_i, p')].$$

So we immediately have

$$
\begin{aligned}
p' &= \sum_{i \in [n]} \Pr_{v_i \sim D_i}[v_i \geq \theta(D_i, p')] \cdot p' && \text{(choice of } p') \\
&= \sum_{i \in [n]} \Pr_{v_i \sim D_i}[v_i \geq \theta(D_i, p')] \cdot \mathbb{E}_{v_i \sim D_i}[v_i \mid v_i \geq \theta(D_i, p')] && \text{(choice of } p', \text{ definition of } \theta(\cdot, \cdot)) \\
&\geq \sum_{i \in A} \Pr_{v_i \sim D_i}[v_i \geq \theta(D_i, p')] \cdot \mathbb{E}_{v_i \sim D_i}[v_i \mid v_i \geq \theta(D_i, p')] \\
&\geq \sum_{i \in A} \alpha_i \cdot \mathbb{E}_{v_i \sim D_i}[v_i \mid v_i \geq \beta_i]. && \text{(monotonicity)}
\end{aligned}
$$

Now consider $B$. For each $i \in B$, by monotonicity and the choice of $B$, we have

$$p' = \mathbb{E}_{v_i \sim D_i}[v_i \mid v_i \geq \theta(D_i, p')] \geq \mathbb{E}_{v_i \sim D_i}[v_i \mid v_i \geq \beta_i].$$

So we have

$$
\begin{aligned}
p' &= \sum_{i \in [n]} \alpha_i \cdot p' && \text{(definition of } \alpha_i) \\
&\geq \sum_{i \in B} \alpha_i \cdot p' \\
&\geq \sum_{i \in B} \alpha_i \cdot \mathbb{E}_{v_i \sim D_i}[v_i \mid v_i \geq \beta_i]. && \text{(monotonicity and choice of } B)
\end{aligned}
$$

Now putting the two parts together, we have

$$
\begin{aligned}
p' &\geq \frac{1}{2}\left(\sum_{i \in A} \alpha_i \cdot \mathbb{E}_{v_i \sim D_i}[v_i \mid v_i \geq \beta_i]\right) + \frac{1}{2}\left(\sum_{i \in B} \alpha_i \cdot \mathbb{E}_{v_i \sim D_i}[v_i \mid v_i \geq \beta_i]\right) \\
&= \frac{1}{2}\sum_{i \in [n]} \alpha_i \cdot \mathbb{E}_{v_i \sim D_i}[v_i \mid v_i \geq \beta_i] \\
&\geq \frac{1}{2} \mathbb{E}_{\{v_i\} \sim \{D_i\}}\left[\max_i v_i\right]. && \text{(definition of } \{\alpha_i\} \text{ and } \{\beta_i\})
\end{aligned}
$$

This concludes the proof. $\qquad \square$

*Proof of Proposition 5.* Let $n$ and $\varepsilon$ be parameters to be fixed later. Let $D_1$ be such that $v_1$ is constantly $n - 2$. Let $D_2, \ldots, D_n$ be identical distributions, where for each $i \in [n] \setminus \{1\}$, $v_i = 1/\varepsilon$ with probability $\varepsilon$, and $v_i = 0$ with probability $1 - \varepsilon$. We let $\varepsilon = o(n^{-2})$. Then the optimal welfare $\mathbb{E}[\max_i v_i]$ is

$$(1 - \varepsilon)^{n-1}(n - 2) + \left(1 - (1 - \varepsilon)^{n-1}\right)/\varepsilon = 2n - 3 + o(1).$$

Then for sufficiently large $n$ and sufficiently small $\varepsilon$,

$$p = \frac{1}{2}\mathbb{E}\left[\max_i v_i\right] = n - \frac{3}{2} + o(1).$$

Observe that $p > n - 2$ for sufficiently large $n$ and sufficiently small $\varepsilon$, so buyer 1 never accepts $p$. As for each $i \in [n] \setminus \{1\}$, the probability that $i$ accepts $p$ is

$$\varepsilon \cdot \frac{1}{\varepsilon}/p = 1/(n - 3/2 + o(1)) = 1/(n-1) + O(1/n^2).$$

So the probability that at least one buyer accepts $p$ is

$$1 - (1 - 1/(n-1) + O(1/n^2))^{n-1} = 1 - \frac{1}{e} + o(1).$$

The claim follows immediately by letting $n \to \infty$ and $\varepsilon \to 0$. $\qquad\square$

*Proof of Proposition 6.* By Theorem 4, there exists some price $p^*$ that extracts total revenue at least

$$\frac{1}{2}\left(1 - \frac{1}{e}\right) \cdot \mathop{\mathbb{E}}_{\{v_i\} \sim \{D_i\}}\left[\max_i v_i\right] \cdot T.$$

Our goal is to compete against this price $p$. We discretize the interval of possible prices $[0, 1]$ into $K$ (to be fixed later) equal pieces, and consider the set of prices $P = \{0, 1/K, \ldots, 1\}$. Observe that there is some price $p' \in P$ satisfying $p' \in [p - 1/K, p]$. Moreover, since the probability that at least one buyer accepts the price is non-increasing in the price, the price $p'$ extracts total revenue at least

$$\frac{1}{2}\left(1 - \frac{1}{e}\right) \cdot \mathop{\mathbb{E}}_{\{v_i\} \sim \{D_i\}}\left[\max_i v_i\right] \cdot T - O(T/K).$$

Now we run any optimal algorithm (e.g., Thompson Sampling [Bubeck and Liu, 2013, Thompson, 1933] or certain versions of UCB [Auer et al., 2002, Lattimore and Szepesvári, 2020]) for finite-armed stochastic bandits with $P$ as the arms. These algorithms achieve regret $O(\sqrt{KT})$ against $p'$, which means the total revenue extracted is at least

$$\frac{1}{2}\left(1 - \frac{1}{e}\right) \cdot \mathop{\mathbb{E}}_{\{v_i\} \sim \{D_i\}}\left[\max_i v_i\right] \cdot T - O(T/K) - O(\sqrt{KT}).$$

Now by choosing $K = T^{1/3}$, the above becomes

$$\frac{1}{2}\left(1 - \frac{1}{e}\right) \cdot \mathop{\mathbb{E}}_{\{v_i\} \sim \{D_i\}}\left[\max_i v_i\right] \cdot T - O(T^{2/3}),$$

as desired. Finally, we remark that the algorithm can be made independent of $T$ by applying the standard doubling trick. $\qquad\square$

*Proof of Proposition 7.* Without loss of generality, suppose the order of arrival is $1, 2, \ldots, n$. For each $i \in [n]$, let $A_i = A_i(v_1, \ldots, v_i) \subseteq [m]$ be the set of items that are available after $i$ leaves. In particular, $A_n$ is the set of items that are not allocated to any buyer. For each item $j \in [m]$, let $q_j$ be the probability that item $j$ is sold, i.e.,

$$q_j = \mathop{\mathrm{Pr}}_{\{v_i\} \sim \{D_i\}}[j \notin A_n].$$

The plan is to lower bound the welfare in two different ways, through the revenue and a lower bound on each buyer's value respectively, and argue that the larger one of the two is at least $1/4$ of the maximum welfare.

First consider the easier part, the revenue. Given the probabilities that the items are sold, the revenue is simply $\sum_j p_j q_j$. This is at least $1/4$ of the maximum welfare if every $q_j \geq 1/2$, which, unfortunately, is not true in general. So we also need to consider the following alternative lower bound of the welfare. For each buyer $i$, fixing $v_i$ and $A_{i-1}$, consider the following feasible (but generally not optimal) set of items to buy.

$$\mathrm{FEA}(v_i, A_{i-1}) = \mathop{\mathrm{argmax}}_{S \subseteq A_{i-1}}\left(v_i(S) - \sum_{j \in S} p_j\right).$$

This is always feasible because it is the utility-maximizing set, and the optimal utility is at least 0 (when buying nothing). So we must always have $v_i(\mathrm{BUY}(v_i, A_{i-1})) \geq v_i(\mathrm{FEA}(v_i, A_{i-1}))$. Now

for each buyer $i$, we can lower bound the expected value of $i$ in the following way (this is structurally similar to the proof of Lemma 3.1 in [Feldman et al., 2014]).

$$\mathop{\mathbb{E}}_{\{v_{i'}\}\sim\{D_{i'}\}}[v_i(\mathrm{BUY}(v_i, A_{i-1}))]$$

$$\geq \mathop{\mathbb{E}}_{\{v_{i'}\}\sim\{D_{i'}\}}[v_i(\mathrm{FEA}(v_i, A_{i-1}))]$$

$$\geq \mathop{\mathbb{E}}_{\{v_{i'}\}\sim\{D_{i'}\}}\left[v_i(\mathrm{FEA}(v_i, A_{i-1})) - \sum_{j\in\mathrm{FEA}(v_i, A_{i-1}(v_1,\ldots,v_{i-1}))} p_j\right] \qquad \text{(prices are nonnegative)}$$

$$\geq \mathop{\mathbb{E}}_{\{v_{i'}\}\sim\{D_{i'}\},\{v'_{i'}\}\sim\{D_{i'}\}}\left[v_i(\mathrm{OPT}_i(v'_1,\ldots,v_i,\ldots,v'_n) \cap A_{i-1}(v_1,\ldots,v_{i-1}))\right.$$
$$\left. - \sum_{j\in\mathrm{OPT}_i(v'_1,\ldots,v_i,\ldots,v'_n)\cap A_{i-1}(v_1,\ldots,v_{i-1})} p_j\right] \qquad \text{(FEA is utility-maximizing)}$$

$$\geq \mathop{\mathbb{E}}_{\{v_{i'}\}\sim\{D_{i'}\},\{v'_{i'}\}\sim\{D_{i'}\}}\left[\sum_{j\in\mathrm{OPT}_i(v'_1,\ldots,v_i,\ldots,v'_n)\cap A_{i-1}(v_1,\ldots,v_{i-1})} a_j(v_i, \mathrm{OPT}_i(v'_1,\ldots,v_i,\ldots,v'_n))\right.$$
$$\left. - \sum_{j\in\mathrm{OPT}_i(v'_1,\ldots,v_i,\ldots,v'_n)\cap A_{i-1}(v_1,\ldots,v_{i-1})} p_j\right] \qquad \text{(Lemma 2)}$$

$$= \sum_j \mathop{\mathbb{E}}_{\{v_{i'}\}\sim\{D_{i'}\},\{v'_{i'}\}\sim\{D_{i'}\}}[\mathbb{I}[j\in\mathrm{OPT}_i(v'_1,\ldots,v_i,\ldots,v'_n)]$$
$$\cdot \mathbb{I}[j\in A_{i-1}(v_1,\ldots,v_{i-1})]\cdot(a_j(v_i, \mathrm{OPT}_i(v'_1,\ldots,v_i,\ldots,v'_n)) - p_j)].$$

Now observe $\mathbb{I}[j\in A_{i-1}(v_1,\ldots,v_{i-1})]$ is independent of everything else in the expectation, so we have

$$\mathop{\mathbb{E}}_{\{v_{i'}\}\sim\{D_{i'}\}}[v_i(\mathrm{BUY}(v_i, A_{i-1}))]$$

$$\geq \sum_j \Pr[j\in A_{i-1}]\cdot\mathop{\mathbb{E}}_{\{v_{i'}\}\sim\{D_{i'}\}}[\mathbb{I}[j\in\mathrm{OPT}_i(v_1,\ldots,v_n)]\cdot(a_j(v_i, \mathrm{OPT}_i(v_1,\ldots,v_n)) - p_j)]$$

$$\geq \sum_j \Pr[j\in A_n]\cdot\mathop{\mathbb{E}}_{\{v_{i'}\}\sim\{D_{i'}\}}[\mathbb{I}[j\in\mathrm{OPT}_i(v_1,\ldots,v_n)]\cdot(a_j(v_i, \mathrm{OPT}_i(v_1,\ldots,v_n)) - p_j)]$$

$$= \sum_j (1-q_j)\cdot\mathop{\mathbb{E}}_{\{v_{i'}\}\sim\{D_{i'}\}}[\mathbb{I}[j\in\mathrm{OPT}_i(v_1,\ldots,v_n)]\cdot(a_j(v_i, \mathrm{OPT}_i(v_1,\ldots,v_n)) - p_j)]$$

$$= \sum_j (1-q_j)\cdot\mathop{\mathbb{E}}_{\{v_{i'}\}\sim\{D_{i'}\}}\left[\mathbb{I}[j\in\mathrm{OPT}_i(v_1,\ldots,v_n)]\cdot\left(\sum_{i'} a_j(v_{i'}, \mathrm{OPT}_{i'}(v_1,\ldots,v_n)) - p_j\right)\right].$$
$$\text{(definition of } a_j)$$

Now summing over $i$, we get

$$\sum_i \mathop{\mathbb{E}}_{\{v_{i'}\}\sim\{D_{i'}\}}[v_i(\mathrm{BUY}(v_i, A_{i-1}))]$$

$$\geq \sum_j (1-q_j)\cdot\mathop{\mathbb{E}}_{\{v_{i'}\}}\left[\sum_i \mathbb{I}[j\in\mathrm{OPT}_i(v_1,\ldots,v_n)]\cdot\left(\sum_{i'} a_j(v_{i'}, \mathrm{OPT}_{i'}(v_1,\ldots,v_n)) - p_j\right)\right]$$

$$= \sum_j (1-q_j)\cdot\mathop{\mathbb{E}}_{\{v_{i'}\}}\left[\sum_{i'} a_j(v_{i'}, \mathrm{OPT}_{i'}(v_1,\ldots,v_n)) - p_j\right]$$

$$= \sum_j (1-q_j)\cdot(2p_j - p_j) = \sum_j (1-q_j)p_j. \qquad \text{(choice of } p_j)$$

Now we put the two bounds together. Recall that the revenue is $\sum_j p_j q_j$, so the welfare is at least

$$
\begin{aligned}
\mathbb{E}_{\{v_i\}\sim\{D_i\}}\left[\sum_i v_i(\mathrm{BUY}(v_i, A_{i-1}))\right] &\geq \max\left\{\sum_j p_j q_j, \sum_j (1-q_j)p_j\right\} \\
&\geq \frac{1}{2}\left(\sum_j p_j q_j + \sum_j (1-q_j)p_j\right) \\
&= \frac{1}{2}\sum_j p_j \\
&= \frac{1}{4}\sum_j \mathbb{E}_{\{v_i\}\sim\{D_i\}}\left[\sum_i a_j(v_i, \mathrm{OPT}_i(v_1,\ldots,v_n))\right] \\
&= \frac{1}{4}\mathbb{E}_{\{v_i\}\sim\{D_i\}}\left[\sum_i v_i(\mathrm{OPT}_i(v_1,\ldots,v_n))\right]. \quad \text{(Lemma 2)}
\end{aligned}
$$

This finishes the proof. $\qquad\square$