# OpenReview forum: "Posted Pricing and Dynamic Prior-independent Mechanisms with Value Maximizers"
_NeurIPS.cc/2022/Conference — NeurIPS 2022 Accept_

### Official Review · Reviewer_Kp2a · 2022-07-10

**Rating:** 6
**Confidence:** 4
**Soundness:** 3 good
**Presentation:** 3 good
**Contribution:** 3 good

**Summary:**

The paper studies the posted price auctions for ROI constrianed value maximizers. Personalized posted prices are shown to be equivalent to posted price auctions for utility maximizers. For anonymous posted prices, a mechanism is provided, proving to be $\frac12(1-1/e)$ approximation. The mechanism is further a to applied prior-independent mechanism with constant approximation, and extended to combinatorial auctions with submodular / XOS agents.

**Questions:**

In the proof of Proposition 4 (Line 193), why the construction satisfy the condition $p\geq \mathbb{E}_{v\sim D}[v]$?

**Limitations:**

Not applicable.

**Strengths And Weaknesses:**

Strengths:
1. The studied problem, auction designs for value maximizers, is important and attracts much attention in the field of auctions recently. The paper is probably one of the first few papers on posted price auctions for value maximizers.
2. Concrete theoretical results are provided. Most proofs, as far as I checked, are corrected.
3. The paper is clearly written and well organized.

Weaknesses:
One proof is not fully justified. See questions.

---

> ### Author Response · Authors · 2022-08-01
> **Author response**
>
> Thank you for your insightful and helpful comments.  Re proof of Proposition 4: this indeed deserves further explanation.  Recall that we require $q_i$ to be in the support of $D_i$ in line 177 (this is without loss of generality, because if $q_i$ is not in the support, we can increase it such that the probability that the buyer accepts stays the same, until $q_i$ is back in the support).  Then we can choose $p_i$ such that $\theta(D_i, p_i) = q_i$ in line 180, and $p_i$ must be unique since we also assume $D_i$ is non-atomic, which also means $\mathbb{E}[v_i \mid v_i \ge q_i] = p_i$.  On the other hand, we know that $\mathbb{E}[v_i \mid v_i \ge x]$ increases monotonically in $x$, and $q_i \ge 0$, so $p_i = \mathbb{E}[v_i \mid v_i \ge q_i] \ge \mathbb{E}[v_i \mid v_i \ge 0] = \mathbb{E}[v_i]$.  We are sorry for the confusion, and will clarify in the paper.

---

> > ### Comment · Reviewer_Kp2a · 2022-08-05
> > **Response to Author's response**
> >
> > Thanks for your response. It addressed my concern.

---

### Official Review · Reviewer_LRn3 · 2022-07-11

**Rating:** 6
**Confidence:** 4
**Soundness:** 3 good
**Presentation:** 4 excellent
**Contribution:** 2 fair

**Summary:**

The paper studies the price of anarchy / efficiency in auctions for ROI-constrained value-maximizing bidders. More specifically it studies posted-price types of auctions. First, it states that an approximation ratio of $\frac{1}{2}$ when non-anonymized posted-prices are possible as it can be reduced to finding posted-prices for utility-maximizing buyers, which is a solve problem. Then, it shows that when using an anonymized posted-price, the approximation ratio is upper-bounded by 0.479. After, it shows that an approximation ratio of $\frac{1}{2}(1-1/e)$ can be obtained with the same allocation as for utility-maximizing buyers. Finally, it provides a $\frac{1}{4}$ approx ratio guarantee when valuations are non-additive.

**Questions:**

I know the paper I'm going to refer to has been published right before NeurIPS submission deadline, so I only put this as a question. Recent results showed that approximation ratios better than $\frac{1}{2}$ can be attained with randomized VCG [4]. Do you think similar ideas could help improve the efficiency of posted-price auctions ?

[4] Auction Design in an Auto-bidding Setting: Randomization Improves Efficiency Beyond VCG. Mehta 2022.

**Limitations:**

I don't have many suggestions on the form of the paper as I find it particularly well-written and easy to read.

I'd say if authors want to "thicken" the content, they can look towards randomized posted-prices as it seems to be the promising direction for such problems.

**Strengths And Weaknesses:**

1. I find Prop. 1 (or 4) is strait-forward: one can write the lagrangian version of the buyer optimization problem between lines 147-148 and observe that choosing $q = \frac{p \lambda^*}{1+\lambda^*}$ where $\lambda^*$ is the optimal lagrange multiplier leads to the optimization problem of a utility-maximizing buyers faced with price $q$. It does not feel like a hugely novel result.

2. I'm not a big fan of Sec. 5. The mechanism has a prior-dependent / non-measurable parameter $p^*$ and the section shows it can be tracked using an adapted sequence of parameters $p_t$ by discretizing its support and running a multi-armed bandit. In my mind, the real challenge about removing prior-dependence is to do so without introducing incentives for the buyers to lie. It's not clear to me that buyers wouldn't be incentivized to lie here, as accepting a price at time $t$ influences the price they get in the future (see [1,2] or [3] for a survey).

3. My main concern about the paper is the strength of the claims. The main results are Th. 3 and 4 in my opinion, but the gap between the upper and the lower bound is large.

[1] Repeated contextual auctions with strategic buyers, Amin et al 2014
[2] Characterizing Truthful Multi-armed Ban- dit Mechanisms, Babaioff et al 2014
[3] Learning in repeated auctions, Nedelec et al 2022

---

> ### Author Response · Authors · 2022-08-01
> **Author response**
>
> Thank you for your detailed and insightful comments.
>
> Re Proposition 1 (and 4): the reviewer is definitely right that that result is not particularly deep in the technical sense, and we are quite frank about it in the paper (we analyze personalized prices as a warm-up, and even say upfront that the proof is "fairly simple").  The proof we provide (which may appear a bit lengthy but definitely not complex) is merely a means to highlight some connections and differences between traditional buyers and value-maximizers.  The argument suggested by the reviewer captures another aspect of the problem (which of course also provides nice insight), and we are happy to discuss that in the paper.
>
> Re "dynamic incentive-compatibility": the reviewer is right that patient buyers may still want to lie in response to that mechanism.  We intended for that mechanism to be a proof of concept.  It is not hard to design a dynamic IC mechanism based on our posted-price mechanisms, as long as buyers are less patient than the seller (which is also a common assumption in dynamic auctions).  For example, one can adopt the exploration-exploitation framework as in [Deng and Zhang, 2021] in the following way: we first run the exploration mechanism by Deng and Zhang for each buyer for a while to learn the approximate value distributions of all buyers.  Then we run our posted-price mechanism with the price slightly lowered to account for potential inaccuracy in the value distributions learned earlier.  By trading off between the lengths of the exploration phase and the exploitation phase, one should be able to get regret $\tilde{O}(T^{2/3})$ against a constant fraction of the optimal revenue.  We will add a remark about this in the paper.
>
> Re "gap between the bounds": while there is a gap between our upper and lower bounds, we note that closing the gap likely involves a fundamentally different pricing scheme, and/or a fundamentally different upper bound argument, from the existing ones in the literature on posted pricing and prophet inequalities.  In particular, most existing pricing schemes are based on variants of the common price $\frac12 \mathbb{E}[\max_i v_i]$.  However, as we show in Proposition 5, this price cannot give a better ratio than $\frac12 (1 - 1/e)$.  Our analysis of the price also deviates significantly from common ones used before.  On the other hand, most existing upper bound arguments are fairly simple, and can be derived by considering instances with only 2 buyers.  However, doing so can never give upper bounds strictly smaller than $1/2$.  In this paper, we already make nontrivial progress and break the barrier of $1/2$.  So in summary, we narrow down the range of the right ratio from the trivial $[0, 0.5]$ to $[0.316, 0.479]$, breaking barriers on both ends of the range.  Pinning down the tight ratio would likely take a whole different paper or two.
>
> Re "randomization may help": we agree randomization in general is a promising direction.  As for the randomized VCG paper in particular, as far as we understand, the techniques there work only when there are only 2 buyers.  We suspect some new ideas are needed to utilize randomization in posted pricing with many buyers.

---

### Official Review · Reviewer_nCDy · 2022-07-11

**Rating:** 7
**Confidence:** 3
**Soundness:** 4 excellent
**Presentation:** 3 good
**Contribution:** 3 good

**Summary:**

The paper studies posted-price auctions for ROI-constrained value
maximizing bidders, who seek to maximize their total value subject to
the condition that the return-on-investment (ROI) on their spend is at
least some prespecified ratio. This is in contrast to
utility-maximizing bidders who seek to maximize their total utility
(value - spend). Past results have shown that in the
utility-maximizing case, using an anonymous posted price mechanism can
achieve a (1/2) approximation of the welfare (or revenue). The current
paper shows that if personalized prices are allowed, the same (1/2)
approximation continues to hold. However, the paper presents instances
where the no anonymous posted price mechanism can achieve an
approximation ratio on welfare greater than 0.483. Furthermore, the
authors show that the using the optimal anonymous posted price
mechanism for the utility-maximizing case obtains a (1/2) (1- 1/e)
approximation of the optimal welfare. The authors also relate their
results to prior-independent dynamic auctions and combinatorial
auctions.


**Questions:**

 For the combinatorial auctions setting, the proof of Proposition~7
uses the factthat the welfare is at least as large as the max of
revenue and buyers' utility. For the welfare to be at least as large
as the revenue, the implicit assumption that the ROI is equal to 1 is
required. Does the same (1/4) guarantee hold for general ROI, or does
the guarantee deteriorate?


**Strengths And Weaknesses:**

The paper extends the results for utility-maximizing bidders to
value-maximizing bidders with ROI constraints, an approach common in
online advertizing. The results are obtained by directly reducing the
analysis to that in the utility-maximizing case using personalized
prices.

The authors construct explicit instances where anonymous posted-prices
achieve strictly less than (1/2) approximation to welfare, though a
tight characterization for this setting is missing.

The authors also provide a tight characterization in the
value-maximizing setting for the usual anonymous posted price
mechanism used in the utility-maximizing setting.

---

> ### Author Response · Authors · 2022-08-01
> **Author response**
>
> Thank you for your insightful and encouraging comments.  Re "target ROI is 1": throughout the paper, we assume each buyer's target ROI ratio is 1.  This is without loss of generality since we consider the liquid welfare (i.e., welfare as measured by each buyer's value adjusted by that buyer's target ROI ratio), which is standard in the study of ROI-constrained value-maximizers (see, e.g., [1-3]).  With the liquid welfare as the benchmark, one can replace the actual value with the adjusted value and rewrite the ROI constraint such that the target ROI ratio is 1.  The liquid welfare is also the maximum amount of money that can be extracted as revenue in the idealized world where all buyers' values are publicly known, which makes it an appropriate benchmark to compare to.  On the other hand, the standard (unadjusted) notion of welfare is less meaningful with value-maximizers, since the full unadjusted welfare can never be extracted as revenue under ROI constraints, even disregarding incentive issues.
>
> [1] Gagan Aggarwal, Ashwinkumar Badanidiyuru, and Aranyak Mehta. Autobidding with constraints.
>
> [2] Santiago Balseiro, Yuan Deng, Jieming Mao, Vahab Mirrokni, and Song Zuo. Robust auction design in the auto-bidding world.
>
> [3] Yuan Deng, Jieming Mao, Vahab Mirrokni, and Song Zuo. Towards efficient auctions in an autobidding world.

---

> > ### Comment · Reviewer_nCDy · 2022-08-07
> > **target ROI**
> >
> > Thanks for the response. Perhaps I was unclear, but it seems to me that the proof of Proposition 7 makes use of the fact that ROI = 1. My question was (1) is this necessary and (2) does the 1/4 bound hold when ROI is not equal to 1 (i.e., prior to rescale/adjustment).  Given your response, it seems to me that the 1/4 guarantee is against the liquid welfare (and not the unadjusted welfare), and so the result holds when ROI \neq 1 as well. Am I understanding this correctly?

---

> > > ### Author Response · Authors · 2022-08-07
> > > **target ROI**
> > >
> > > Yes, that is correct (sorry for the confusion).  Please let us know if you have any other questions or comments.

---

### Meta-Review · Area_Chair_2kTG · 2022-08-20

**Recommendation:** Accept
**Confidence:** Less certain

**Metareview:**

This paper got uniformly positive reviews. That said, reading into the actual text of the reviews, it is evident that the results are not as strong as one might like. The biggest limitation is that the result rely heavily on personalized pricing for reducing the problem to one of utility maximization for utility maximizers; posted prices are often not a very attractive approach in practice.

**Award:**

No

---

### Decision · Program_Chairs · 2022-09-14

Accept